# Molecular Characterisation of Uterine Endometrial Proteins during Early Stages of Pregnancy in Pigs by MALDI TOF/TOF

**DOI:** 10.3390/ijms22136720

**Published:** 2021-06-23

**Authors:** Dorota Pierzchała, Kamila Liput, Agnieszka Korwin-Kossakowska, Magdalena Ogłuszka, Ewa Poławska, Agata Nawrocka, Paweł Urbański, Aleksandra Ciepłoch, Edyta Juszczuk-Kubiak, Adam Lepczyński, Brygida Ślaska, Krzysztof Kowal, Marinus F. W. te Pas, Magdalena Śmiech, Paweł Leszczyński, Hiroaki Taniguchi, Leyland Fraser, Przemysław Sobiech, Mateusz Sachajko, Magdalena Herudzinska, Chandra S. Pareek, Mariusz Pierzchała

**Affiliations:** 1Department of Animal Genomics and Biodiversity, Institute of Genetics and Animal Biotechnology, Polish Academy of Sciences, Jastrzebiec, ul. Postepu 36A, 05-552 Magdalenka, Poland; pierzchalada@gmail.com (D.P.); k.stepanow@igbzpan.pl (K.L.); a.kossakowska@igbzpan.pl (A.K.-K.); m.ogluszka@igbzpan.pl (M.O.); e.polawska@igbzpan.pl (E.P.); a.nawrocka@ighbzpan.pl (A.N.); p.urbanski@igbzpan.pl (P.U.); a.cieploch@gmail.com (A.C.); 2Laboratory of Biotechnology and Molecular Engineering, Department of Microbiology, Prof. Wacław Dąbrowski Institute of Agricultural and Food Biotechnology, State Research Institute, Rakowiecka 36 Street, 02-532 Warsaw, Poland; edyta.juszczuk-kubiak@ibprs.pl; 3Department of Physiology, Cytobiology and Proteomics, West Pomeranian University of Technology, ul. K. Janickiego 32, 71-270 Szczecin, Poland; adam.lepczynski@zut.edu.pl; 4Faculty of Animal Sciences and Bioeconomy, Institute of Biological Bases of Animal Production, University of Life Sciences in Lublin, ul. Akademicka 13, 20-950 Lublin, Poland; brygida.slaska@up.lublin.pl (B.Ś.); krzysztof.kowal@up.lublin.pl (K.K.); 5The Netherlands Wageningen UR Livestock Research, Animal Breeding and Genomics, P.O. Box 9101, 6700 HB Wageningen, The Netherlands; marinus.tepas@wur.nl; 6Department of Experimental Embryology, Institute of Genetics and Animal Breeding, Polish Academy of Sciences, Jastrzebiec, ul. Postepu 36A, 05-552 Magdalenka, Poland; m.smiech@igbzpan.pl (M.Ś.); p.leszczynski@igbzpan.pl (P.L.); h.taniguchi@igbzpan.pl (H.T.); 7Department of Animal Biochemistry and Biotechnology, Faculty of Animal Bio-Engineering, University of Warmia and Mazury in Olsztyn, ul. Oczapowskiego 5, 10-719 Olsztyn, Poland; fraser@uwm.edu.pl; 8Internal Disease Unit, Department of Clinical Sciences, Faculty of Veterinary Medicine, University of Warmia and Mazury in Olsztyn, 10-719 Olsztyn, Poland; psobiech@uwm.edu.pl; 9Faculty of Biological and Veterinary Sciences, Institute of Veterinary Medicine, Nicolaus Copernicus University, ul. Gagarina 7, 87-100 Toruń, Poland; mateuszsachajko@gmail.com (M.S.); mherudzinska@umk.pl (M.H.); pareekcs@umk.pl (C.S.P.); 10Centre for Modern Interdisciplinary Technologies, Nicolaus Copernicus University, ul. Wilenska 4, 87-100 Toruń, Poland

**Keywords:** differentially expressed proteins, pregnancy, endometrium, implantation, proteome, Polish Large White, pigs, MALDI TOF/TOF, biomarkers

## Abstract

The molecular mechanism underlying embryonic implantation is vital to understand the correct communications between endometrium and developing conceptus during early stages of pregnancy. This study’s objective was to determine molecular changes in the uterine endometrial proteome during the preimplantation and peri-implantation between 9 days (9D), 12 days (12D), and 16 days (16D) of pregnant Polish Large White (PLW) gilts. 2DE-MALDI-TOF/TOF and ClueGO^TM^ approaches were employed to analyse the biological networks and molecular changes in porcine endometrial proteome during maternal recognition of pregnancy. A total of sixteen differentially expressed proteins (DEPs) were identified using 2-DE gels and MALDI-TOF/TOF mass spectrometry. Comparison between 9D and 12D of pregnancy identified APOA1, CAPZB, LDHB, CCT5, ANXA4, CFB, TTR upregulated DEPs, and ANXA5, SMS downregulated DEPs. Comparison between 9D and 16D of pregnancy identified HP, APOA1, ACTB, CCT5, ANXA4, CFB upregulated DEPs and ANXA5, SMS, LDHB, ACTR3, HP, ENO3, OAT downregulated DEPs. However, a comparison between 12D and 16D of pregnancy identified HP, ACTB upregulated DEPs, and CRYM, ANXA4, ANXA5, CAPZB, LDHB, ACTR3, CCT5, ENO3, OAT, TTR down-regulated DEPs. Outcomes of this study revealed key proteins and their interactions with metabolic pathways involved in the recognition and establishment of early pregnancy in PLW gilts.

## 1. Introduction

The endometrium layers experience functional cyclical changes regulated by ovarian hormones, growth factors, and cytokines [1]. In recent studies, physiological and molecular examinations of the uterine endometrium during the peri-implantation period have revealed several genes associated with signalling pathways involved in embryo-uterine crosstalk to be differentially expressed [2,3,4,5,6],. The present paper hypothesized that during the peri-implantation period, the dynamic proteome alterations in the uterine fluid could be visible between the 10D and 14D of pregnancy [7,8]. In the first few weeks of pregnancy in the pig, these critical alterations could be defined as the preimplantation period after fertilisation, that is, early pregnancy before maternal recognition of pregnancy (gestation period duration of 1–10 days: in this study the 9D); the peri-implantation period of maternal recognition of pregnancy (gestation period duration of 11–13 days: in this study the 12D) and, the peri-implantation period to determine establishment or failure of pregnancy (gestation period duration of 14–19 days: in this study the 16D) [3]. The proteome profile of the endometrium remains in a close relationship with the uterus condition during the embryo implantation process [9]. The proper development of embryos depended on this initial pregnancy period, that is, between the 9D and 16D after fertilisation. The molecular mechanism of embryo implantation and pregnancy maintenance has been shown to require the synergistic action of many factors [10]. Besides progesterone (P4) produced by the corpus luteum, estradiol derived from the embryo is also required that stimulates maternal recognition of pregnancy and, also for early pregnancy maintenance [11]. In the pig, at least four fully functional embryos must be present to sustain multifetal pregnancy [3]. The estimated mortality rate of the embryo in the peri-implantation period in swine is about 20–30%, which means that every third fertilised oocyte degenerates before day 35 of pregnancy. The most critical moment of gestation for embryonic survival is the first three weeks (20 days) of gestation. Early embryonic loss can occur during pre-elongation development on 10D (day 10), trophoblastic elongation on 12D, or attachment of conceptus trophoblast to endometrium on 16D. Embryonic loss on the ten days is mainly associated with asynchrony during embryo uterine development. On the 12D, the embryos compete for the best possible space in the uterus, which is essential for access to nutrients. Conceptus trophoblast attachment on 16D decides embryonic survival [12].

The course of pregnancy has a direct impact on the effectiveness of pig breeding, and it depends on many genetic and environmental factors, that is, uterine capacity, or the placenta’s efficiency [13]. This multifactorial dependency caused that molecular mechanisms involved in proper pregnancy stabilisation in the early embryonic loss have still not yet been fully explained. Genetic determinants and anomalies of the embryo as the main causes of pregnancy disorders [14], but are not fully responsible for several cases of miscarriages [15]. The large proportions of embryos lost have been significantly related to the mother’s endocrine system’s malfunction and pregnancy disorders [16,17]. Pre-eclampsia (PE) disease of early pregnancy is an important reason for perinatal morbidity and mortality and ranges from 2% to 8% of pregnancies [18].

Identification of protein profile at early stages of pregnancy allows a better characterisation of molecular processes and biological pathways in pigs. A previous study reported the presence of twenty-one proteins that were uniquely expressed on the 12–13 and 15–16 days of pregnancy [19]. Furthermore, a systemic analysis of identified proteins revealed cell adhesion and cytoskeletal organisation as two of the major functions, which are important for the establishment and maintenance of pregnancy [19]. In another study [20], alterations of protein abundances were identified in annexin A4, beta-actin, apolipoprotein, ceruloplasmin and afamin haptoglobin, prolyl-4-hydroxylase, aldose-reductase and transthyretin proteins and were associated with progression of the estrous cycle and pregnancy from the 9D to 12D using MALDI-TOF/TOF [20]. Protein profiling studies could also help to discover more specific biomarkers for high and low-risk pregnancies to define surveillance in pregnancy and administer interventions [21].One of the most frequent causes of pregnancy disorder is the insufficient blood supply [22]. Insufficient uterine blood supply results in a lack of nutrients to the embryo, which could have an unfavorable effect on the embryo development due to superficial implantation nourished mainly by histotroph [23]. So far, most studies on the molecular characterisation of global protein profiles in the early stages of pregnancy have been performed on embryos [24,25]. Proteomic profile studies in human provided a profound knowledge of the endometrial fluid proteomic changes during the secretory phase of the menstrual cycle and the implantation window [26]. Distinct expression patterns have been detected between early and mid-pregnancy and between viable and arresting conceptus attachment sites [27]. Moreover, due to the multifactorial nature of uterine endometrial dysfunction resulting in infertility, it seems necessary to examine the combination of multiple molecular markers for a better characterisation of endometrial receptivity. The limited data on protein expression profiles of endometrial tissues in the literature has provided a comprehensive characterisation of the pig uterine endometrium at the early stages of pregnancy. Therefore, this study aimed to determine the molecular mechanism and biological processes influencing pathways that convey receptivity in the pig endometrium. For this purpose, we performed the identification and characterisation of the endometrial protein profiles on 9D, 12D, and 16D of pregnancy in the Polish Large White (PLW) gilts using matrix-assisted laser desorption ionisation time of flight mass spectrometry (MALDI TOF/TOF MS) approach.

## 2. Results

### 2.1. Identification of Uterine Endometrial DEPs during the Preimplantation and Peri-Implantation Period

The overall protein profiles were evaluated by two-dimensional electrophoresis (2DE) with precise protein identification by mass spectrometry—MALDI TOF/TOF MS (Figure 1). Such an approach was known for a long time and is still widely used in the quantitative analysis of protein identification and variations of protein levels. Identification of proteins with different expression patterns between the subsequent pregnancy stages was based on the bioinformatics software, the Bruker Biotools ™, ProteinScape ™ and Mascot ™, (Matrix Science) server platform, and evaluation of peptides mass fingerprint concerning Uniprot and SwissProt databases. 

In our study, we performed three comparisons (12D vs. 9D; 16D vs. 9D, and 16D vs. 12D) among the early stages of pregnancy and identified sixteen differentially expressed proteins (DEPs) in the uterine endometrium during the preimplantation (9D) and peri-implantation period (12D and 16D) in pregnant gilts (Table 1). 

Comparison between 9D vs. 12D of early stages of pregnancy identified upregulated APOA1, CAPZB, LDHB, CCT5, ANXA4, CFB and TTR (*p* < 0.05) DEPs and downregulated ANXA5 and SMS (*p* < 0.05) DEPs. While, the comparison between 9D vs. 16D of pregnancy identified upregulated HP, APOA1, ACTB, CCT5, ANXA4, and CFB (*p* < 0.05) DEPs and downregulated ANXA5, SMS, LDHB (*p* < 0.01), ACTR3, HP, ENO3, and OAT (*p* < 0.05) DEPs. However, the comparison between 12D vs. 16D of pregnancy identified upregulated HP and ACTB (*p* < 0.05) DEPs and down-regulated CRYM, ANXA4 (*p* < 0.01), ANXA5, CAPZB, LDHB, ACTR3, CCT5, ENO3, OAT, and TTR (*p* < 0.05) DEPs. (Figure 2 and Figure 3).

### 2.2. Network Interaction of Identified DEPs during the Preimplantation and Peri-Implantation Period and Their Involvement in the Biological Pathways and Molecular Processes

We used the Cytoscape 3.8 and ClueGo 2.5.5 software for biological network visualisation and data integration. Based on the fold of enrichment analysis, selected DEPs allowed to establish protein interaction networks within specific pathways and molecular processes (Figure 4, Figure 5 and Figure 6). Comparing the preimplantation (9D) and peri-implantation period (12D) of pregnancy using the pathway’s enrichment significance analysis revealed the interaction of ACTR3, HP, ACTB, AOPA1, CAPZB proteins network in the vesicle-mediated transport pathway, (CAPZB, ACTB, APOA1, ANXA5 proteins network in the Haemostasis pathway, ACTR3, CFB, HP, ACTB, TTR proteins network in the innate immune system pathway, and TTR, AOPA1, SMS, OAT, LDHB, ENO3, CRYM proteins network in the metabolism pathways (Figure 4). While, comparing the preimplantation (9D) and peri-implantation period (16D) of pregnancy using the pathway’s enrichment significance analysis revealed the interaction of ANXA4 and ANXA 5 proteins network in the lipase inhibitor activity pathway and TTR, CRYM, APOA1, and HP proteins network in thyroid hormone transport pathway (Figure 5). However, comparing the peri-implantation periods of 12D and 16D pregnancies using the pathway’s enrichment significance analysis revealed the interaction of SMS, OAT, ENO3 and LDHB proteins network in the arginine and proline metabolism pathways (Figure 6).

Cytoscape ClueGo analysis further revealed the involvement of DEPs in various molecular processes (Appendix A and Figure 6) and differentially associated gene ontology terms (Figure 7) involving in lipase inhibitor activity (ANXA4, ANXA5, and APOA1), prostaglandin synthesis (ANXA4, ANXA5), binding and uptake scavenger receptor (HP, APOA1) metabolism of fat-soluble vitamins, retinoid metabolism and transport amyloid precursor proteins from ordered fibrils, (TTR, APOA1). Moreover, obtained results showed that some proteins are playing an important role in urea cycle pathways (LDHB, OAT), glycolysis and gluconeogenesis (LDHB, ENO3), cysteine, methionine metabolism (LDHB, SMS), arginine, proline, and urea cycle metabolism (OAT, SMS).

### 2.3. Validation of Identified Uterine Endometrial DEPs during the Preimplantation and Peri-Implantation Period

To validate our DEPs results, ANXA4 and HP proteins were selected for further analysis using western blot (Figure 8 and Appendix A). Similar to the proteomics data, the western blot validation experiment indicated that ANXA4 and HP abundance increased in pregnant endometrium during 9D to 16D of early pregnancy. Statistical analysis of the validation experiment (Figure 9) revealed that changes in the protein levels of ANXA4 and HP proteins relative to the GAPDH loading control were statistically significant (*p* ≤ 0.05).

## 3. Discussion

### 3.1. Molecular Characterisation of Upregulated DEPs during 9D, 12D and 16D of Pregnancy

In our study, we investigated the molecular mechanism and biological processes influencing pathways that convey receptivity in the pig endometrium during early stages of pregnancy. Recently several studies confirmed that dynamic communication between embryo and endometrium is the known fundamental for embryo implantation during porcine pregnancy [1,6,17,19,20]. In our study, using the MALDI TOF/TOF MS approach we characterized the endometrial protein profiles on 9D, 12D, and 16D of pregnancy in the Polish Large White (PLW) gilts, and finally, 16 DEPs were identified. We observed a significant increase of apolipoprotein A1 (APOA1) on 12D and 16D of pregnancy compared to 9D. This protein is one of the main transporters of essential lipids to the developing conceptus. The APOA1 is a primary acceptor for cholesterol in extrahepatic tissues and also has potent anti-inflammatory properties. It may also be involved in embryo attachment and the regulation of cytokine production [28]. APOA1 associated with lipid metabolism has been detected in the uterine fluid of pregnant and cyclic pigs [25], cattle [29], and sheep [30]. Jalali et al. [20] noted increasing amounts of APOA1 between day D and 13D in both pregnant and cyclic sows. Pregnancy-associated upregulation of APOA1 was reported at the transcriptome level in the horse, whereas it was downregulated in the pregnant mares’ uterine fluid [31]. The presence of APOA1 in the uterine fluid of pregnant and cyclic pigs, cattle and sheep demonstrated its contribution to the embryo maternal dialogue [28,30]. Moreover, APOA1 expression is dysregulated in pregnancy disorders. Previous studies have shown increased expression of APOA1 in secretory endometrium in pelvic endometriosis. Additionally, the study by Verma et al. [32] suggested that low APOA1 expression is critical in establishing pregnancy and elevated APOA1 expression in chorionic villi correlates with early miscarriage (EM). While the study of Brosens et al. suggested APOA1 as a potent anti-inflammatory molecule and a critical marker in human infertility [33].

In our study, we noted an increase of chaperonin containing TCP1, subunit 5 (CCT5) protein on 12D of pregnancy compared to 9D and 16D. Moreover, we observed a higher level of this protein on 16D of pregnancy than 9D. Similar changes were observed in the studies of the profile of proteins involved in embryo maternal interaction and the signalling of maternal recognition of pregnancy in the horse [31]. Furthermore, CCT5 was found among foetal genes in early pregnancy to be significantly downregulated in pregnancy lost compared to a healthy pregnancy [34]. The chaperonin containing TCP-1 is necessary for folding newly synthesised proteins, including actin and tubulin [35,36].

The complement system is vital for the development of the placenta and, consequently, for the foetus [37]. CFB is a component of a proteolytic cleavage cascade related to protecting the cell from damage caused by organisms like microbes and is also involved in the clearance of damaged cells and cellular debris [38]. Our study results showed a significant increase of complement factor B (CFB), on 12D and 16D compared to 9D of pregnancy. The substantial change in the abundance of CFB on 12D of pregnancy compared to cycling endometria was also reported by Moza Jalali et al. [39]. It is known that CFB has a physiologically important function in reproductive traits such as uterine epithelium growth [40]. The CBF gene was mapped in the region with several QTLs and specific genes linked with reproductive traits of sows such as uterine capacity and ovulation rate, and litter size [41]. Buske et al. [42] also showed a significant association of pig litter size with the CBF gene’s polymorphism.

Our results provide evidence for a significant increase in the endometrium of the uterine annexin A4 (ANXA4) level on 12D compared to 9D and 16D of pregnancy in pigs. ANXA4 may be necessary for regulating ion and water transport across the endometrial epithelium [43]. Moreover, ANXA4 has been shown to play a role in the early phases of apoptosis, with translocation from the nucleus to the cytosol during this process. Therefore, it might be involved in the endometrial apoptosis during the middle and late secretory phase of the menstrual cycle and play a crucial role in the receptive process [44]. Decreases of the ANXA4 level may result in placental structural abnormality and dysfunction in human foetuses [45]. Annexins have been suggested to promote cellular adhesion, an essential step required for blastocysts to adhere to the endometrium [46]. Elevated ANXA4 mRNA correlated with an increasing level of progesterone was reported by Talbi et al. [47]. Numerous microarray studies of the human endometrium have shown that the ANXA4 transcript is significantly upregulated during the menstrual cycles secretory phase compared with the proliferative phase [48,49]. In the pig, ANXA4 was differentially expressed with the oestrous cycle and pregnancy progression from 9D to 12D [20]. Also, larger quantities of ANXA4 in the oviductal fluid of pregnant mares compared to cyclic mares have been reported by Smits et al. [31]. It is considered that ANXA4 can be an important biomarker for polarised epithelial cells, which could be implicated in the regulation of calcium-activated epithelial chloride channels [49].

In our results, we detected a significantly higher level of the transthyretin (TTR) on 12D of pregnancy in comparison to 9D and 16D. The TTR is one of the significant proteins which concentration increases have been observed in uterine flushing at implantation in the mouse [50]. An increase of TTR level in the endometrium of pregnant pigs during the peri-implantation period could be associated with its role as an important serum protein carrier of thyroid hormones and retinol, critical for implantation and development of the mammal’s embryo [51]. Thyroxine is a crucial thyroid hormone responsible for the rate of cellular metabolism stimulating oxidation processes in cells [52]. The pivotal function of TTR was also confirmed in humans’ studies where it plays a role in uterus receptivity and normal pregnancy. TTR is abundant in the uterine cavity, uterine secretion, placenta, and serum of pregnant females in the peri-implantation uterus and the first trimester of pregnancy [53,54]. For this reason, the increase of TTR protein expression in the uterine horn observed on 12D indicated that TTR could be one of the important indicators of uterine receptivity in pigs, as it is in humans.

Our study noted a significant increase of the β Actin (ACTB) abundance on 16D compared to 12D or 9D of pregnancy. Remodelling actin fibres is critical for the cell membrane’s morphological organisation, for the generation of cell-to-cell interactions, and cellular adhesion to the extracellular matrix. During early pregnancy, the endometrium undergoes extensive structural modification and growth, which is essential for accommodation and support rapid embryo development, and changes in the actin cytoskeletal elements are expected [55]. ACTB is the predominant form responsible for cell migration and gene expression regulating the cell cycle and cell migration [56]. All these processes are essential for the plasticity of tissues, especially the endometrium. The cytoplasmic actin is also crucial for regulating the endometrium’s rearrangement during pregnancy [57,58]. These results proved why ACTB is a poor reference gene for PCR and Westerns of porcine endometrial tissues.

Our study showed a higher abundance of haptoglobin-like (HP) on 16D in pregnant sows than in 12D and 9D. This protein is commonly recognised as an ‘acute phase’ protein, strongly binding to haemoglobin. Various cytokines are exchanged between the mother and the embryo during pregnancy, stored, and resubmitted again to the mother [59]. Haptoglobin is a protein that might regulate this process. Haptoglobin is secreted not only by hepatic cells but also in the lung, adipocytes [60], and the endometrium [61]. Whereas the embryo does not express haptoglobin mRNA, it could accumulate substantial amounts of maternal haptoglobin within the embryonic coats and blastocyst fluid [62]. Besides its haemoglobin binding function, haptoglobin could reduce leukocyte activity by competitive binding to CD11b/CD18 [63]. Since haptoglobin reduces adhesion and invasion of the leukocytes to extracellular matrices [61], it may serve within the embryonic coats as early protection against maternal immune cells [64]. Increased levels of haptoglobin during trophoblast invasion may further induce immunological tolerance. These results were consistent with the findings of Bottini et al. [65].

### 3.2. Molecular Characterisation of Down-Regulated DEPs during 9D, 12D and 16D of Pregnancy

The present research showed a significant and systematic decrease in the level of annexin A5 (ANXA5) on 12D and 16D of early pregnancy compared to 9D. This protein has previously been isolated from the human placenta [66], what could be related to that ANXA5 is described as a potent anticoagulant and act as an inhibitor of phospholipid dependent blood coagulation reactions [67,68]. Other research showed that plasma levels of ANXA5 remain low and stable during pregnancy and after delivery [69]. Studies by Degrelle et al. [70] showed that, during elongation and the morphological transition of the conceptus, there is an increase in ANXA5. ANXA5 is involved in the cellular cytoskeletal architecture, morphogenesis of the embryo. Such observation was obtained in the analysis of pig conceptus elongation, which involved an alteration in trophectoderm cellular structure, the development of filopodia, and the underlying endoderm movement [71]. Moreover, the comparative characterisation of four lipocortin family members confirmed that ANXA5 acts as a placental anticoagulant protein and is necessary for maintaining placental integrity [72,73,74,75]. Shu et al. [76] detected the decreased expression of ANXA5 on trophoblasts of pre-eclamptic placentas correlated with a rise in markers for blood coagulation. Overall, this would facilitate the tolerance of the immunologically “foreign” conceptus while allowing activation of the immune system to defend against microbial and viral infections.

Another down-regulated protein of particular interest was L lactate dehydrogenase B chain (LDHB), for which a significantly lower level was noted on 12D and 16D compared to 9D of pregnancy. It has been reported that the embryo implantation process is connected with the enhanced glucose catabolism leading to the accumulation of lactic acid in the extracellular microenvironment, and this has become a recognised metabolic feature of mammalian blastocysts [77]. Therefore, a large amount of lactic acid may contribute to the formation of an acidic environment in early pregnancy, promoting the embryo’s successful implantation [78]. Xiao et al. [79] showed that mouse uterus tissue’s acidification was associated with early uterine preparation for embryo implantation. At a late stage of the blastocyst and during the early stages of implantation, there is a shift in lactate dehydrogenase (LDH) isoforms from LDHB promoting pyruvate formation to LDHA promoting lactate formation [80]. Furthermore, lactate from the blastocyst can modulate the immune response of the mother [81,82] through the decrease the function of cytotoxic T cells, and increase the production of the angiogenic factor VEGF (vascular endothelial growth factor) by macrophages, which dominate the implantation site [83,84]. It is suggested that the region of high lactate/low pH created by the blastocyst modulates the local immune response activity, helping to develop immune tolerance [85]. Therefore, in our study, a significant increase of LDHB could be associated with the production of lactate by the blastocyst to facilitate implantation, activation of signalling to facilitate angiogenesis, and immune modulation to prevent maternal rejection.

We also observed a significant decrease in the level of actin-related protein 3 (ACTR3) on 16D compared to 12D or 9D of pregnancy. A higher level of ACTR3 protein could be associated with its function, essential in preimplantation embryos [86,87]. Studies of the Arp2/3 complex expression showed that specific mRNA transcript was identified in oocytes and embryos in preimplantation porcine embryos. The high expression was noted at the four cells stage and then gradually decreased through to the blastocyst stage. Synthesis of Arp2 was detected to rising from one cell to four cells then significantly decreased at morula stages through the blastocyst stages [88]. The actin cytoskeleton is essential for conceptus elongation as constricted regions along the filamentous conceptuses contain polarised trophoblast cells with a distinct F actin array. Focal adhesions are macromolecular complexes comprised of heterodimeric transmembrane integrin receptors that connect ECM to the actin cytoskeleton to regulate cell growth, proliferation, survival, migration, gene expression, and cell morphology [89]. Endometrial morphology is dynamically regulated by altering the levels of steroid hormones throughout the menstrual cycle or pregnancy. Cyclic structural modifications observed in the endometrium are caused by the interaction of many cellular processes, including the cytoskeleton’s rearrangement. The remodelling of actin fibres is crucial for the cell membrane’s morphological organisation, generation of intercellular interactions, and cell adhesion to the extracellular matrix, and all these processes determine endometrium tissue plasticity. Control of cytoplasmic actin is particularly critical for regulating endometrial rearrangement during pregnancy [56,57]. Both proteins (ACT and ACTR3) are related to RHO GTPases Activating WASPs and WAVEs pathways. WAVE complex signals downstream of activated Rac to stimulate F actin polymerisation in response to immune receptor signalling. ACT and ACT3 are regulated differently during gestation [90].

Our research showed a significant decrease in β-enolase (ENO3) on 16D compared to 12D or 9D of pregnancy. Higher expression of ENO3 was noted in the uterine luminal fluid of pregnant heifers on 16D of pregnancy and was associated with glycolysis/gluconeogenesis pathways [29]. ENO3 was found two-fold upregulated gene among genes in microarray studies in pregnant women myometrium [91]. ENO3 is a glycolytic enzyme involved in energy metabolism. A similar result was also noticed for ornithine aminotransferase (OAT); with a significant decrease of OAT level on 16D compared to 12D or 9D of pregnancy. It is known that OAT is involved in the formation of proline from the amino acid ornithine [92]. Ornithine is decarboxylated by ornithine decarboxylase (ODC1) to produce putrescine, which is the substrate for spermidine and spermine production. Putrescine, spermidine, and spermine are critical regulators of proliferation, migration, and differentiation of trophectoderm cells during pregnancy peri-implantation period [92,93]. In our study, the noted decrease in the OAT level on 16D of pregnancy can be caused by the increased demand for ornithine to produce putrescine for embryo development. The mRNA expressions of the OAT and ODC1 were detected in the human endometrium in the secretory phase, which suggests that cell proliferation and collagen synthesis might be augmented by increased arginase activity in the endometrium [94].

Proteomic analysis showed a decrease in spermine synthase (SMS) level on 12D and 16 compared to 9D of pregnancy. A study by Jalali et al. [20] also reported a decrease in the SMS level in the endometrium of sows from 9D to 12D of pregnancy. Spermine plays an important role in angiogenesis [95]. Extensive post-attachment vascular remodelling occurs in the porcine endometrium between 12 to 15 days of gestation what could be connected with increased nutrient requirements for the growth of conceptus [96]. Spermine synthase plays a role in early mammalian embryogenesis, placental trophoblast growth, and embryonic development in the uterus. It was down-regulated in the placenta of intrauterine growth restricted (IUGR) pig foetuses on the 60th day of gestation [97]. Our results also showed a significantly lower haptoglobin precursor (HP) level on 16D compared to 9D. The observed decrease of haptoglobin protein precursor could be associated with an increase of haptoglobin. It may be hypothesised that endometrial cells tend to accumulate haptoglobin precursor. Haptoglobin precursor was observed in the human endometrium in the proliferative and secretory phase, with a two-fold increase in the secretory phase [98,99]. It is postulated that haptoglobin, based on its immunosuppressive properties, could be involved in protecting the foetus from the maternal immune response [100].

Ketimine reductase mu crystallin (CRYM), showed a significantly lower level on 16D of pregnancy than on 12D. This protein is involved in regulating the free intracellular concentration of triiodothyronine and access to its nuclear receptors [101]. CRYM importance during the first half of pregnancy could be associated with maternal thyroid hormone influence because maternal thyroid failure would lead to several pregnancy complications, including pre-eclampsia, premature labour, foetal death, and low birth weight and intellectual impairment in the offspring in human [102,103]. We observed a decrease in the level of F actin capping protein subunit beta (CAPZB) on 16D compared to 12D or 9. This protein plays a role in regulating cell morphology and cytoskeletal organisation [104]. CAPZB was founded to be down-regulated in the conception cycle compared to the non-conception cycle in the human endometrial fluid aspirate [105]. In livestock, functional analysis of proteomic profiles during the early stage of pregnancy was reported in cows [106], sows [25] as well as in sheep [107]. In sows, it has been reported that above 47% of histotrophic proteins show differential expression between 10D and 13D pregnancies [25]. During the implantation period, in cows, 34 proteins with a different expression were primarily associated with rebuilding the uterus environment in preparation for implantation, embryo growth and development, and the uterine immune response [106]. Similarly, in sheep, MALDI TOF/TOF analysis revealed 19 proteins overexpressed on 16D of pregnancy, associated with remodelling of regulation of the immune system, stabilisations of oxidative stress, and adaptation to increasing requirements for the nutrition elements of growing embryos [107]. In humans’ case, a receptive phase showed elevated levels of proteins linked to the mother’s immune system [99]. In pigs, the fertility rates are generally very high, but the early embryonic loss during the second and third weeks of gestation may affect the potential litter size [20]. Therefore, the quantity changes of these proteins might be considered for further investigation as potential biomarkers useful to improve pig breeding traits. Our proteomic analysis provided a unique insight into further understanding of the biological mechanisms underlying pig endometrial receptivity.

### 3.3. Protein Networks Interaction—Molecular Processes

Our results showed the involvement of DEPs in various molecular processes (Appendix A), including the lipase inhibitor activity (ANXA4, ANXA5 and APOA1), prostaglandin synthesis (ANXA4 ANXA5), binding and uptake scavenger receptor (HP, APOA1). In pregnancy recognition, one of the key factors was the innate immunomodulation response, and the scavenger receptors have been demonstrated to play an important role in innate immune defence by acting as pattern recognition receptors (PRR) [108,109]. We also found the molecular processes as the metabolism of fat-soluble vitamins, retinoid metabolism and transport (TTR, APOA1) (Figure 3). In the female, the effect of vitamin A deficiency on the reproductive outcome depends upon the time when deficiency is imposed and its severity causing reproduction to fail before implantation [110]. Moreover, our study demonstrated the potential influence of DEPs on methionine cysteine and methionine metabolism (LDHB, SMS). Methionine is an essential amino acid, but the importance of methionine in embryonic development and the underlying molecular mechanisms. The study by Cai et al. [111] described the function and mechanism of methionine in improving embryo development and implantation. Other results strongly indicate that cysteine or glutathione can improve porcine ICSI derived embryos’ developmental competence by reducing intracellular ROS level and the apoptosis index [112]. Our studies revealed that amyloid precursor proteins molecular processes would be affected by identified differentially expressed proteins (TTR, APOA1). In line with the literature, it was known that communications between the amyloid precursor protein and notch could be important for the implantation process as it was shown that the expression of notch 1 increased during the implantation window [113,114]. In our study, identified DEPs were involved in the extracellular matrix organisation (β actin; ACTR3), actin cytoskeleton organisation (ACTB, CAPZB), cellular lipid metabolism (APOA1, TTR), early phases of apoptosis (ANXA4), maternal immunomodulation (HP), angiogenesis (LDHB), and thyroid hormones secretion (CRYM; TTR). These proteins could be essential for modulating endometrial receptivity for conceptus attachment and embryo development in the early pregnancy stage (Figure 2, Figure 3 and Figure 4).

## 4. Materials and Methods

The research material consists of twelve gilts of PLW breed. After finding the second oestrus gilts were inseminated twice and slaughtered at 9D (*n* = 4), 12D (*n* = 4) and 16D (*n* = 4) of early pregnancy. The pregnancies were confirmed by the presence of the typical characteristic for the number of given embryos determined by the number of corpora lutea. Searching and elution of embryos were stopped after getting up to 12 embryos from each PLW gilt on an average. The endometrial tissue slices were collected from the middle part of the mesometrial side of the uterus horn (all areas of the endometrium lumen) as described previously [3]. Collected samples of the endometrium for the proteomic analysis were frozen in liquid nitrogen and stored at minus 80 °C. All experiments were conducted under the national guidelines for agricultural animal care in Poland. All procedures involving animals were approved by the third local animal ethics committee’s decision in Poland no. 70/2006.

### 4.1. Protein Extraction

A 100 mg of liquid nitrogen precooled samples were homogenised using a ceramic mortar and pestle and transferred into lysis buffer (7 M urea, 2 M thiourea, 4% *w*/*v* CHAPS) with protease inhibitor cocktail (mix of AEBSF, aprotinin, bestatin, E64, leupeptin, and pepstatin A) (Thermo Fisher^TM^, Grand Island, NY, USA). Lysates were incubated for 10 min on ice and then centrifuged with 12,000× *g* for 25 min at 4 °C. Next, supernatants containing proteins were collected. Isoelectric focusing (IEF) was performed on Protean i12^®^ IEF Cell (Bio-Rad Laboratories, Inc., Hercules, CA, USA), after measuring total protein concentration using Bradford assay (Bio-Rad Protein Assay). Nonlinear 3–10, 17 cm ReadyStrip™ IPG Strips (Bio-Rad Laboratories, Inc.) were rehydrated with buffer (7 M urea, 2 M thiourea, 4% *w*/*v* CHAPS, 1% *w*/*v* DTT, 2% *v*/*v* BioLyte^®^ 3/10 Ampholyte) containing 600 µg of protein in 300 µL. The IEF was conducted in a total of 90,000 Vh. For SDS-PAGE IPG strips were reduced for 15 min in equilibration buffer (6 M urea, 0.5 M Tris-HCl pH 6.8, 2% *w*/*v* SDS, 30% *w*/*v* glycerol) with 1% DTT and then alkylated for 20 min in the same buffer containing 0.5% iodoacetamide instead of DTT. SDS-PAGE was conducted in a PROTEAN^®^ II xiCell electrophoretic chamber (Bio-Rad Laboratories, Inc.) on 12% polyacrylamide gels in TGS buffer (25 mM Tris-HCl, 192 mM glycine, 0.1% SDS) under the following conditions: 40 V for 2.5 h and then 100 V for 16.5 h, both steps with cooling. Subsequently, two-dimensional polyacrylamide gels with separated proteins were visualised with colloidal Coomassie Brilliant Blue G-250 according to the modified method of [115] and [116]. The gels were scanned with Gel Doc XR+ Gel Documentation System (Bio-Rad).

### 4.2. Protein Identification

Images of protein spots from 2D-IEF electrophoresis were analysed with PDQest advanced v. 8.0.1 software to identify proteins with differing expression. Experiment normalisation was performed according to the local regression model (LOESS). The different expressions of proteins were measured as the densitometric quantity of spots and were estimated using the Wilcoxon rank test. Selected protein spots with statistically different quantities (Figure 1) were excised from 2-D gels manually, decolourised, dehydrated, vacuum dried, and digested overnight at 37 °C using the Trypsin Profile IDG kit (Sigma Aldrich, Darmstadt, Germany) according to the manufacturer’s instructions. Peptide samples were placed on the MALDI target plate and covered with α cyano-4-hydroxycinnamic acid (5 mg/mL in 50% ACN with 0.1% TFA) in a 1:1 ratio. Digested proteins were analysed with MALDI TOF/TOF mass spectrometer, ultrafleXtreme™ (Bruker Daltonics, Billerica, MA, USA). Mass spectra were collected, calibrated (Peptide Mass Standard II, mass range 700–3200 Da, Bruker Daltonics), and analysed in Flex Analysis 3.4 and Biotools 3.2. software. The peptide mass spectra were collected by accumulating analyses from 2000 laser shots. Peptide mass fingerprint (PMF) specific for particular spots were compared with the mammalian proteomic database (SWISS-PROT) using the MASCOT software as a search engine with the following settings: one missing tryptic cleavage allowed, 150 ppm mass accuracy, carbamidomethylation of cysteine as a fixed modification, and methionine oxidation as a variable modification. The results were validated based on the score of the protein sequence coverage obtained in the MASCOT software. 3.3 Bioinformatics protein interaction analysis.

Biological processes interaction network was elaborated on the base the fold of enrichment analysis using Cytoscape 3.8 TM software and ClueGO v2.5.5, with the following selection criteria: (1) Statistical test used the enrichment/depletion (Two-sided hypergeometric test) using *p*-value cut-off = 0.05000000074505806; (2) Correction method using Bonferroni step down with minimum GO level 2; maximum GO level 6; GO group equal to true; kappa score threshold equal to 0.4; (3) Organism analysed was homo sapiens using genes in WikiPathways, KEGG pathways, REACTOME reactions and REACTOME pathways. [117,118].

### 4.3. Validation of Endometrial Proteins Using Western Blotting

Frozen endometrial tissues from 9D (*n* = 4), 12D (*n* = 4) and 16D (*n* = 3) of early pregnancy were grinded using a mortar and a pestle in liquid nitrogen. Protein extraction was performed using RIPA buffer containing 25 mM Tris-HCl pH 7.6, 150 mM NaCl, 1%NP-40, 1% sodium deoxycholate, 0.1% SDS (Thermo Fisher Scientific Inc.) and Protease Inhibitors (complete™, Mini Protease Inhibitor Cocktail, Roche Diagnostics GmbH, Basel, Switzerland). The samples were placed in an ultrasonic cleaner with ice for 5 min, then manually ground by a plastic pestle. Next, the samples were centrifuged at 12,000× *g* at 4 °C for 15 min. The supernatant was collected. 40 μg endometrial proteins were mixed with 5× Laemmli Sample Buffer (62.5 mM Tris-HCl pH 6.8, 20% glycerol, 2% SDS, 5% β-mercaptoethanol, bromophenol blue) and boiled at 95 °C for 5 min. Proteins were separated by 12% SDS-PAGE (40 V for 30 min, 100 V for 2.5 h) and transferred to a PVDF membrane (100 mA for 1.5 h). 5% non-fat dry milk in TBST (25 mM Tris, 150 mM NaCl, 0.05% *(v*/*v)* Tween20, pH = 7.7) was used as a blocking buffer for 1 h at room temperature and incubated with primary antibodies overnight at 4 °C. Specific antibodies are listed in Table 2. Blots were stripped using stripping buffer (0.2 M glycine, 0.1% *(w*/*v)* SDS, 1% *(v*/*v)* Tween20, pH 2.2) twice for 10 min each, washed in PBS twice and washed in TBST twice.

The chemiluminescence signals were detected using Clarity Western ECL Substrate (Bio-Rad Laboratories, Inc.) and ChemiDoc XRS+ Gel Imaging System (Bio-Rad Laboratories, Inc.). The band lane’s intensity value was quantified using ImageJ (Rasband, W.S., ImageJ, U. S. National Institutes of Health, Bethesda, MD, USA, https://imagej.nih.gov/ij/index.html, accessed on 1 March 2021). The Relative density of each band was calculated by dividing by the standard samples (mixture of all samples) for each blot to account for variations between multiple blots. The adjusted intensity values of target proteins (annexin A4 and haptoglobin) were calFculated by dividing the relative density of each sample lane (ANXA4, HP) by the relative density of the loading-control (GAPDH) for the same lane.

## 5. Conclusions

Uterine embryo implantation is a multistep, multifaceted process requiring precisely timed molecular mechanisms covering several divergent molecular mechanisms and biological processes. We showed that using proteomics techniques enables us to find upregulated (APOA1, CAPZB, LDHB, CCT5, ANXA4, CFB, TTR, HP, and ACTB,) and downregulated (ANXA5, SMS, LDHB, ACTR3, HP, ENO3, OAT, CRYM, ANXA4, CAPZB, CCT5, and TTR) proteins involved in very diverse molecular processes and pathways such as, vesicle-mediated transport, hemostasis, arginine and proline metabolism, innate immune system, lipase inhibitor activity, and thyroid hormone transport, respectively. Present proteomics studies provide more information about how early pregnancy is maintained on a molecular level in a large litter size animal like the pig as a potential animal model for early human pregnancy. Furthermore, it is valuable because the law prohibits human embryos’ cultivation beyond 14D, and preventing research into this and subsequent events, such as gastrulation, when an overall plan of building the body is being established. Due to the inability to study implantation in humans, animal models are necessary to decipher this process’s molecular and mechanical events. It is known that different species exhibit different mechanisms by which embryonic implantation takes place. Therefore, different animal species may be more suitable as models for specific steps in the human implantation process. For example, pigs and sheep are potential candidates for the early stages of implantation because they have extended adhesion and attachment phases. New molecular techniques for measuring changes in the embryo and endometrium before and during the embryonic implantation provide a better understanding of pregnancy development’s critical moment. However, these approaches still require detailed in-depth research. Therefore, it is essential to develop standardised research methods during the early stage of embryo development and pregnancy to understand the molecular basis of the physiology of the uterine endometrium.

## Figures and Tables

**Figure 1 ijms-22-06720-f001:**
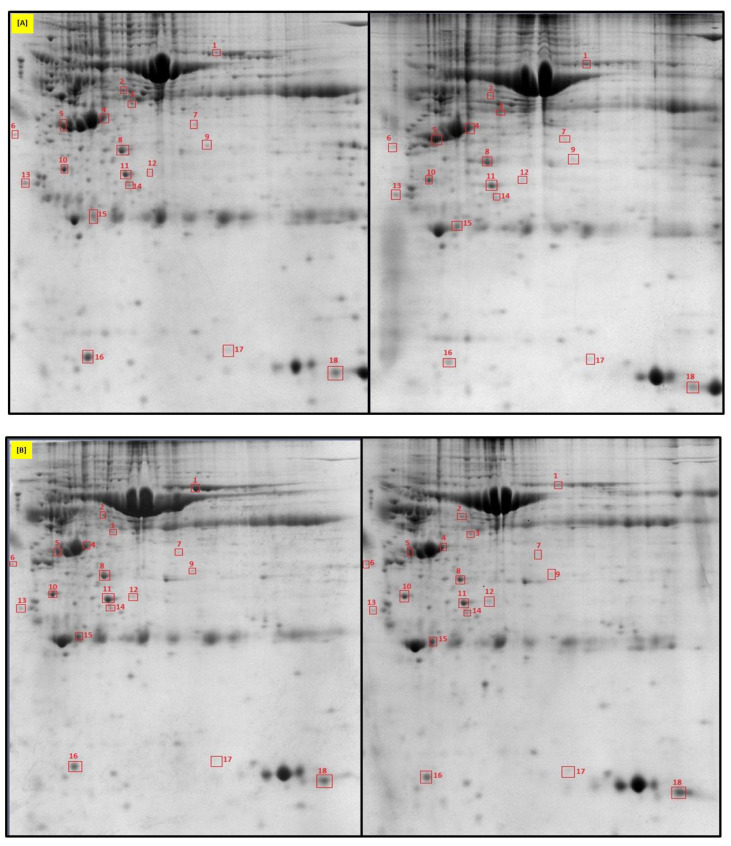
Identification of eighteen differntially expressed endometrial proteins at 9D (**A**), 12D (**B**), and 16D (**C**) of early pregnancy in Polish Large White gilts using two-dimensional IEF electrophoresis gels and MALDI TOF/TOF mass spectrometer.

**Figure 2 ijms-22-06720-f002:**
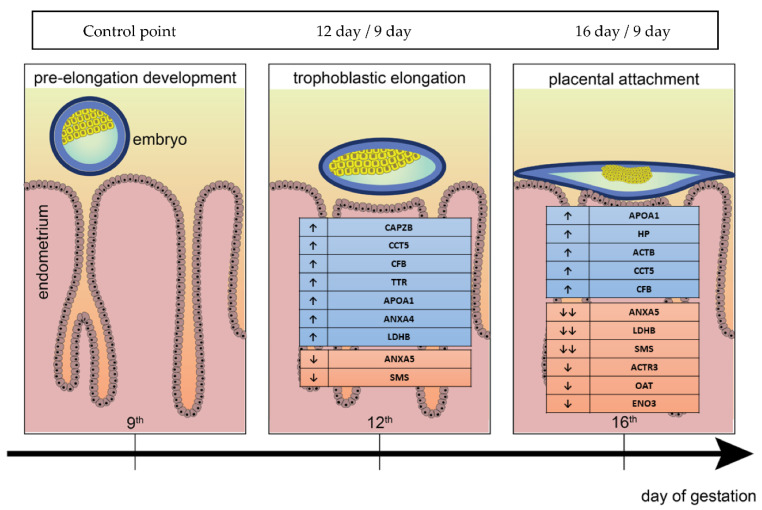
Graphical visualisation of identified DEPs between the preimplantation and peri-implantation periods using two-dimensional electrophoresis (2DE) coupled with MALDI TOF/TOF. Arrow indicating the upregulated and downregulated altered protein abundance. ↑—increase protein expression in 12th or 16th day in comparison to 9th day; ↓—decrease protein expression in 12th or 16th day in comparison to 9th day; down arrows (→) means time line of stages of early pregnancy in PLW gilts.

**Figure 3 ijms-22-06720-f003:**
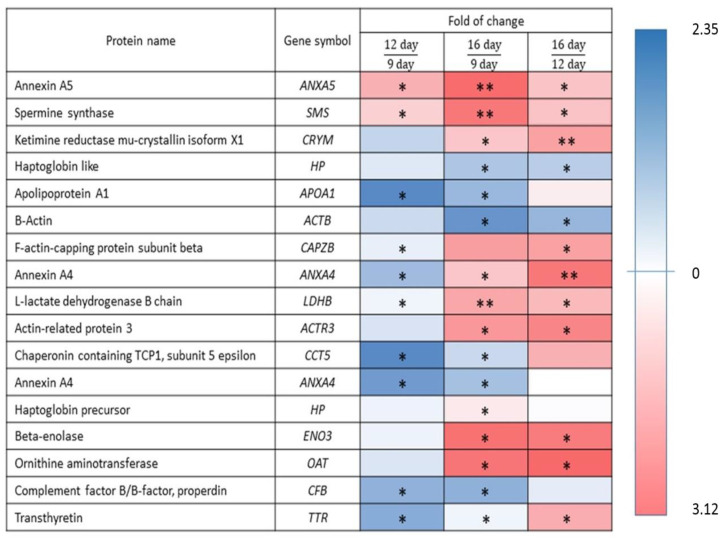
Effects of the fold of change (FC) observed in the DEPs of endometrial tissue during 9D, 12D and 16D of pregnancy. *—*p* < 0.05; **—*p* < 0.01; (Fold change marked in blue colour indicate upregulated (↑)DEPs and fold change marked in red colour indicated downregulated (↓)DEPs.

**Figure 4 ijms-22-06720-f004:**
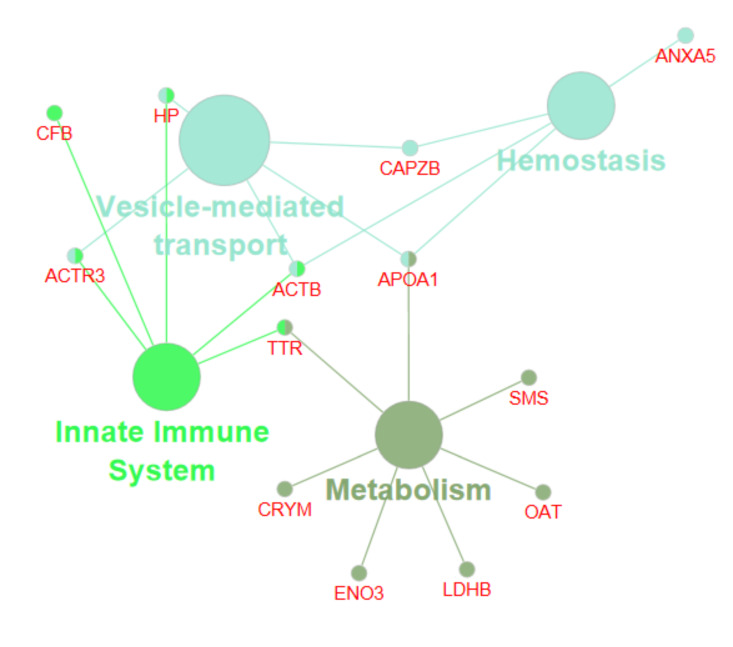
Visualisation of the functionally grouped gene ontology and protein interaction networks of DEPs in porcine endometrium. Comparison of the preimplantation (9D) and peri-implantation period (12D) of pregnancy based on pathway’s enrichment significance analysis by Cytoscape ClueGo^TM^. The nodes’ size reflects the pathway’s enrichment significance.

**Figure 5 ijms-22-06720-f005:**
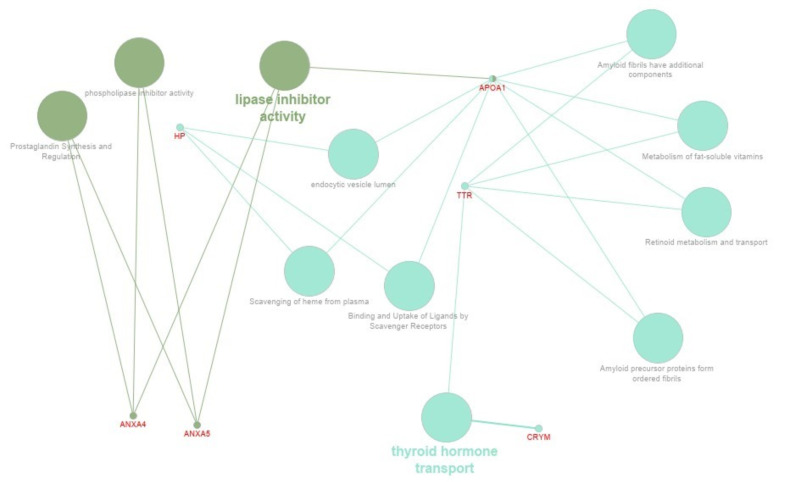
Visualisation of the functionally grouped gene ontology and protein interaction networks of DEPs in porcine endometrium. Comparison of the preimplantation (9D) and peri-implantation period (16D) of pregnancy based on pathway’s enrichment significance analysis by Cytoscape ClueGo^TM^. The nodes’ size reflects the pathway’s enrichment significance.

**Figure 6 ijms-22-06720-f006:**
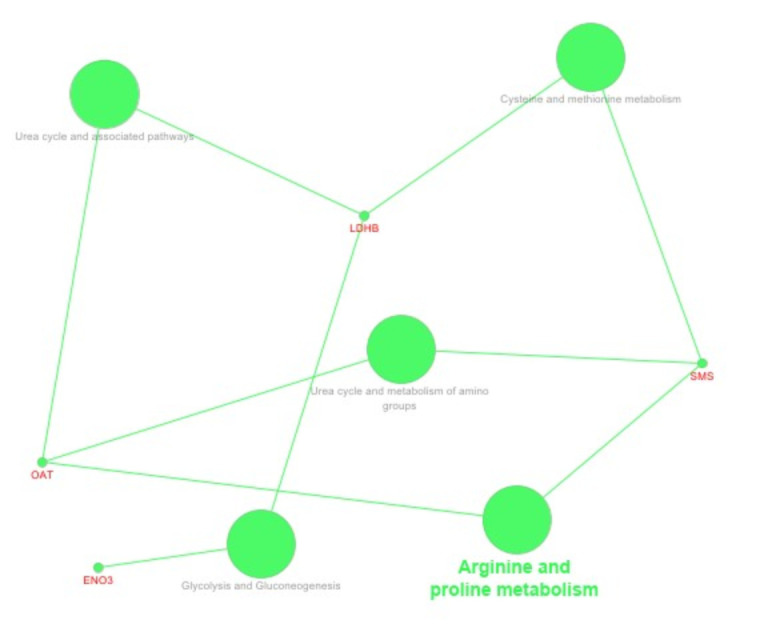
Visualisation of the functionally grouped gene ontology and protein interaction networks of DEPs in porcine endometrium. Comparison of the peri-implantation periods of (12D and 16D) pregnancy based on pathway’s enrichment significance analysis by Cytoscape ClueGo^TM^. The nodes’ size reflects the pathway’s enrichment significance.

**Figure 7 ijms-22-06720-f007:**
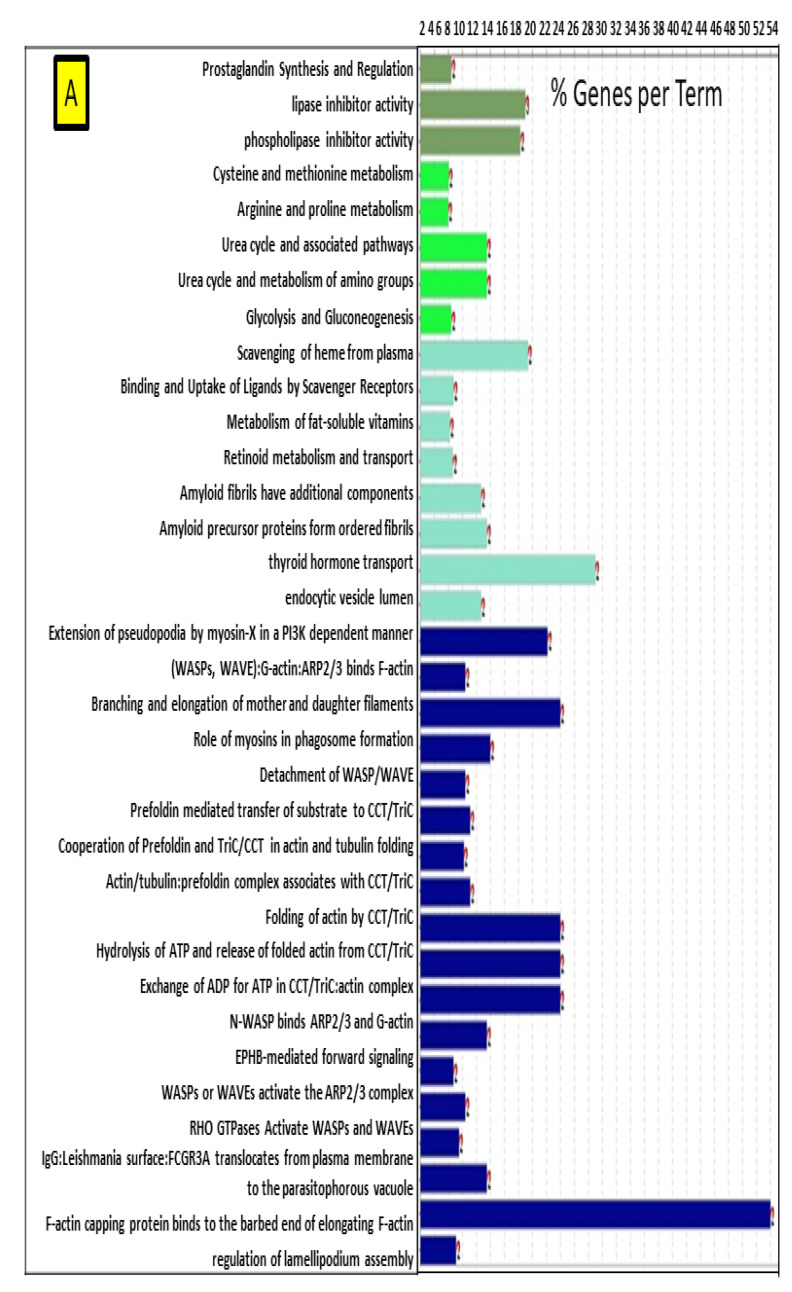
Identification of GO term organisation associated with DEPs using Cytoscape ClueGO^TM^. The differentially associated gene ontology terms were shown with different colours. The dark green denoted lipase inhibitor activity: The bright green denoted arginine and proline metabolism: The blue-green denoted thyroid hormone transport. The blue denoted the F-actin pathway. Terms within each group were coloured similarly. The number and percentage of contributing genes for each term can be inferred in the bar diagram 6 (**A**) and pie diagram 6 (**B**). The asterisk indicates the significance of the related terms and groups based on corrected *p*-values. The *p*-value < 0.001 was assigned with ** and Kappa score equal to 0.43 rotate labels in figure.

**Figure 8 ijms-22-06720-f008:**
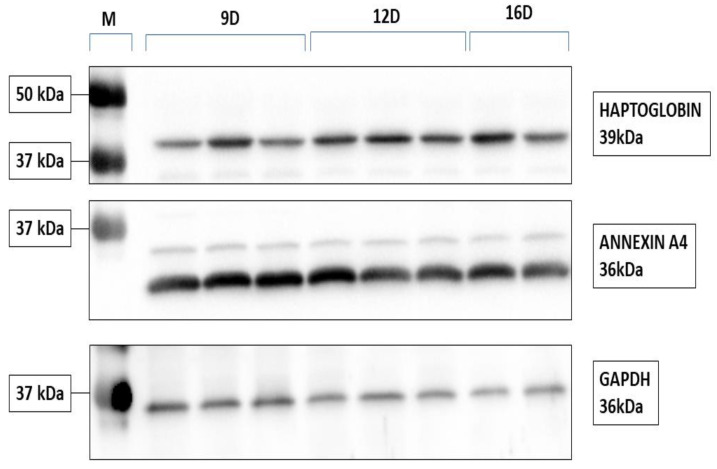
Representative validation of annexin A4 and haptoglobin by 1D western blot in the porcine endometrium at 9D, 12D, and 16D of early pregnancy in PLW gilts.

**Figure 9 ijms-22-06720-f009:**
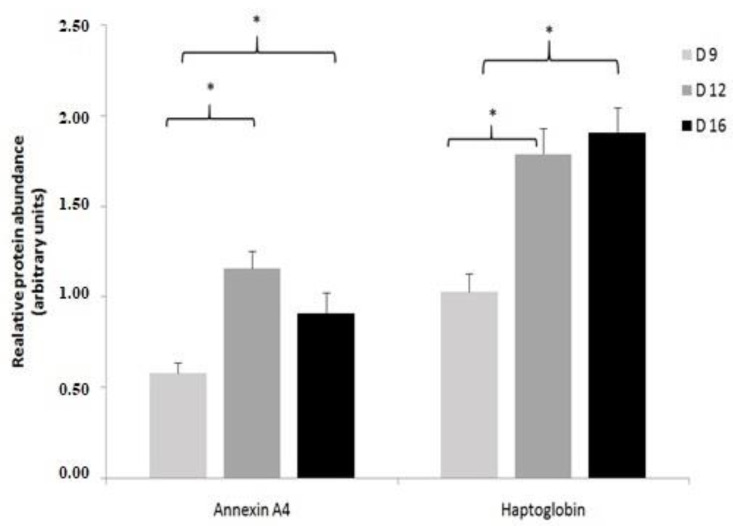
The statistical summary of the protein abundance analysis of four biological repeats relative to the loading control (GAPDH). Data are presented as mean ± SEM, * *p* ≤ 0.05; Wilcoxon test. (For more details: Appendix A).

**Table 1 ijms-22-06720-t001:** Characteristic of identified proteins differentially expressed in uterine endometrial tissue during 9D, 12D, and 16D of pregnancy in PLW gilts.

Protein Name	Gene Symbol	Protein Spot Order Number on 2D Gel	Molecular Weight (Da)	Theoretical pH	Seq Coverage %	Score	Protein Function	NCBI Identification Code	NCBI Ref. Seq. Number
Annexin A5	ANXA5	1	36.148	4.94	37%	88	Blood coagulation hemostasis	gi|335293906	XP_003129266.2
Spermine synthase (PABs-domain containing protein)	SMS	2	41.239	4.87	41%	83	Polyamine biosynthesis	gi|350595577	XP_020935312.1
Ketimine reductase mu crystalline isoform X1	CRYM	3	33.508	5.16	14	54	Transcription co-repressor activity	gi|335284508	XP_003124605.2
Haptoglobin like	HP (L)	12	38.481	6.51	11	50	Acute phase response, defense response	gi|350596912	ACD93463.1
Apolipoprotein A1	APOA1	4	30.325	5.48	33	47	Cholesterol metabolism, transport	gi|164359	AAA30992.1
B Actin	ACTB	5	41.737	5.29	34	61	Cytoskeleton organization	gi|45269029	AAS55927.1
F actin capping protein subunit beta	CAPZB	10	33.017	5.53	25	83	Isomerase, apoptotic cell clearance	gi|147899312	NP_001090923.1
Annexin A4	ANXA4	9	35.829	5.7	52	120	Promotes membrane fusion	gi|264681432	NP_001161111.1
L-Lactate dehydrogenase B chain	LDHB	8	36.612	5.57	32	82	Glycolysis, carbohydrate metabolism and oxido reduction	gi|164518958	NP_001106758.1
Actin related protein 3	ACTR3	7	47.425	5.61	19	40	Maintenance of cell polarity, asymmetric cell division	gi|197251946	NP_001127815.1
Chaperonin containing TCP1, subunit 5 epsilon	CCT5	6	59.641	5.57	22	64	Cochaperone, protein folding	gi|523580068	NP_001265708.1
Annexin A41	ANXA41	11	35.829	5.7	31	68	Promotes membrane fusion	gi|264681432	NP_001161111.1
Haptoglobin precursor	HP (P)	16	13.843	6	23	56	Acute phase response, defense response	gi|47522826	NP_999165.1
Beta enolase	ENO3	15	47.130	8.05	12	79	Regulation of blood coagulation	gi|113205498	NP_001037992.1
Ornithine oxo-acid aminotransferase	OAT	14	48.477	6.44	18	41	Catalytic, transaminase activity	gi|297591979	NP_001172070.1
Complement factor B/B factor, properdin	CFB	13	85.865	7.45	13	46	Innate immunity, complement activity	gi|156120152	NP_001095294.1
Transthyretin	TTR	17	16.081	6.29	12	22	Transport, extracellular matrix organization	gi|975233	CAA61120.1

**Table 2 ijms-22-06720-t002:** List of specific immunoblot antibodies utilized in the western blotting validation experiment.

Target Protein	UniProtKB Accession Number	Gene Symbol	Molecular Weight	Dilution	Host Species	Conjugate	Company	Code Number
Primary antibody								
GAPDH	P00355	GAPDH	36 kDa	1:10,000	rabbit	polyclonal	Abcam plc (Cambridge, UK)	ab190304
haptoglobin	Q61646	HP	39 kDa	1:500	rabbit	polyclonal	Abcam plc	ab231976
annexinA4	P08132	ANXA4	36 kDa	1:500	rabbit	polyclonal	Abcam plc	Ab232805
Secondary antibody								
Peroxidase AffiniPure Goat Anti-Rabbit IgG (H+L)				1:10,000	goat	polyclonal, horseradish peroxidase (HRP) conjugates	Jackson ImmunoResearch Europe Ltd. (Ely, UK)	111-035-003

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
