# Peer review of "Molecular Characterisation of Uterine Endometrial Proteins during Early Stages of Pregnancy in Pigs by MALDI TOF/TOF"

_ijms, 2021, doi:10.3390/ijms22136720_

Round 1
Reviewer 1 Report
Pierzchala et al examine endometrial proteomic differences between days 9, 12 and 16 of pregnancy in gilts. They identify 16 proteins whose abundance changes between these timepoints. The study is of interest, though is very similar to other recent publications and I am concerned by the lack of appropriate controls and validation.
Significant edits to English (e.g. neither the first nor last sentence of the abstract make sense), and grammar (e.g. (d, 12D and 16D is not separated by commas, nor has D been defined), is required.
Specific comments
Line 46/47: It would be useful to delineate between up and downregulated DEPs, and also define ‘several’ as a specific number.
Line 60-63: This sentence is difficult to understand as written. Rewrite as ‘Physiological and molecular examination of the uterine endometrium during the per-implantation period has revealed several genes associated with signalling pathways involved in embryo-uterine crosstalk to be differentially expressed’. There are a significant number of other instances where the English used makes it difficult to understand the intended statement, changes in tense, misuse of words (e.g. ‘caused’ on line 86) or there are spelling issues (e.g. line 64 ‘genes’ should be ‘gene’). The use of hyphens throughout the title and the manuscript is distracting (e.g. endome-trium on line 80, en-vironment’s on line 93) when this should be left to the editorial team.
The authors describe the identification of 16 DEPs, of which 9 are changed between day 9 and 12, and 13 between days 12 and 16. It is not clear why the authors compare day 9 and 16. How do the authors know that these proteins relate to differences between pre-implantation and post-implantation specifically, rather than changes in abundance that might normally occur around these times independent of the presence of an embryo? The absence of equivalent non-pregnant controls means that there is very little to conclude from the data provided. This is exemplified by the identification of APOA1, which is observed in both pregnant and cycling pigs between these time points – so this protein is not specifically a difference between pre and post implantation events.
As 4 viable embryos are required for ongoing pregnancy, did the authors exclude samples in which there was evidence of ongoing embryonic loss (i.e. did they confirm that there were at least 4 viable embryos when endometrial sampling took place, and no pre-elongation, elongation or implantation losses) that may impact on proteomic abundances? If there was variation across samples in embryo numbers, how was this accounted for in their analysis, particularly if it was not confirmed whether there were further embryos present after 12 had been flushed?
Does their p-value (which is greater than 0.05 – why?) account for multiple comparisons (i.e. day 9 vs 12, 9 vs 16 and 12 vs 16)?
Line 137-138: ANXA4 is mentioned twice. Similarly, on lines 140 and 141 – ANXA4 is mentioned as both an upregulated and a downregulated protein between day 9 and 16, and on lines 143 and 144 between days 12 and 16. It is noted in Figure 1 that these have been identified from different spots – this should be clearly noted within the text. Are these different isoforms?
Similarly, in Table 1: HP appears twice – first as Haptoglobin like and then as haptoglobin precursor. If these are in fact 2 different proteins then they need to be better delineated with the use of an L (for like) or P (precursor), or the words themselves to aid the reader.
The authors need to perform westerns to confirm protein identities.
Figure 1: do the arrows indicating altered protein abundance reflect changes relative to the preceding timepoint (i.e. arrows on day 12 are relative to day 9, and those on day 16 are relative to day 12)? This should be clarified in the legend.
Figure 2 – this is a table, not a figure. Further, a fold change of -0.32 would not be biologically significant. Only those proteins with an abundance change of >1.5 or >-1.5 fold should be considered.
Line 162: the authors note that DEPs were classified into three main biological pathways. However, in Figure 3 alone there are 4 pathways/processes. More simply, Figures 3-5 represent protein interaction networks that change between each timepoint.
Line 180-181: It is stated that ‘These DEPs were also important for immune modulation, preventing maternal rejection and further proper embryo implantation and development.’ Would the authors please clarify how this was determined.
Line 186-188: Interpretation should be limited to the discussion.
Lines 178-196: The authors should refer to the relevant protein interaction network (Figures 3-5). E.g. for lines 178-180 = Figure 3. For lines 193-196 = Figure 5.
Table 2: What is the P value for each of these GO term associations? I would expect that the identification of only 2-3 proteins within each pathway does not result in significance. Table 2 and Figure 6 do not significantly differ in the information being presented – one should be deleted.
Figure 6 – define what each colour represents. It is noted in the legend on line 201 that ‘terms within each group were coloured similarly’ but it is unclear what these groups relate to/how they have been classified – biological processes I presume, but using what analysis?
The discussion is unnecessarily broken up into individual results, with each identified protein as its own subsection. This makes it difficult to find the ‘take home messages’ (i.e. key findings). The authors should discuss a selection of relevant pathways/biological processes (e.g. metabolism, immune system etc) and their significance – mentioning identified proteins only in passing within this broader context and allow the reader to explore specific proteins if they wish to do so.
The authors note several proteins identified match those from Jalali et al (2015), a carbon copy experiment (albeit with controls). Similar work has also been published by Kolakowska et al 2017, yet this is not referenced.
Line 470: briefly describe how second oestrus was ‘found’ (determined).
Line 528: Why was Homo sapiens used for the pig?
Conclusions: beyond the pig being a potential model – what are the overarching conclusions to take away from the current study – what is the significance of the proteins identified overall for embryo-endometrial crosstalk/pregnancy recognition, did the authors identify novel proteins and what further could be done to investigate their importance?
Minor comments:
Author names should be written with surname last, or with commas between the surname and first name for easier identification.
Line 46: ‘Results data revealed’ – Results or data, not both.
Line 49 (and elsewhere in the manuscript): change ‘of level’ to ‘in the level of’.
Line 51: ‘Obtained results concluded’ does not make sense.
Line 221: the study of Verma et al should read the study by Verma et al.
Use of the word ‘noticed’ (e.g. line 284) should be changed to ‘noted’.
Author Response
attachment file-1

Reviewer 2 Report
The present study was dedicated to examine the changes in the global proteome of the porcine endometrium during the peri-implantation period, days 9, 12, and 16. Authors detected seventeen proteins with altered endometrial expression between studied days and concluded that these results may be important for understanding molecular changes related to pregnancy establishment in the pig.
Aim of the study is interesting but similar studies have been previously performed and already published: Chae et al. (2011, Proteome Sci), Jalali et al. (2015, J Proteomics; 2016, Mol Reprod Dev) and Kolakowska et al. (2017; Reprod Fertil Dev). Thus, presented results are not novel and Authors did not manage to make them more interesting or of high importance. There is no validation of the obtained results with Western blot or immunostaining techniques. Moreover, the data are descriptive and Authors (22 co-authors) do not attempt to provide some mechanistic studies to examine possible function any of differentially expressed proteins.
Introduction does not contain a clear justification why this study has been performed. Especially what is new in regard to previously published results. Include a research hypothesis. Try to underlie the novelty.
I strongly do not agree with the statement provided in lines 542-548 (Conclusions), that pig may be a model to examine implantation for human as implantation and placentation in these species differ substantially with non-invasive epitheliochorial placenta in pigs and invasive hemochorial in human.
Sample collection on day 16: on this day porcine conceptuses adhere to the endometrium; please, specify whether endometrial samples were collected from implantation sites or inter-implantation sites. In fact, Authors do not provide information how conceptuses were obtained to confirm pregnancy: flushed from uterine horn or uterine horn was open longitudinally first?
There are several words in this manuscript written with a dash, like “devel-opment”, “endo-metrium”, “exam-ine” and many more.
Author Response
attachment file 2

Round 2
Reviewer 1 Report
The authors have revised the manuscript, taking into account a number of reviewer comments. The authors have included the necessary western blot validation data, which enhances the manuscript substantially. However, they agree with my comment that, based on their current experimental design, it can not be determined whether the observed changes relate to differences between pre-implantation and post-implantation, or rather changes in abundance that might normally occur around these times independent of the presence of an embryo. This is a significant flaw that remains unaddressed. Given the dynamic communication between the embryo and endometrium - the authors are unable to conclude whether the changes observed even reflect a normal time course (without an embryo).
Author Response
Response to Reviewer-1: In the revised manuscript, we have done extensive editing of the English language and style as per the IJMS guidelines to the authors. As Indicated by the reviewer, there is still some flaw that remains unaddressed. In the revised manuscript, we have tried our best to address the majority of critics and suggestions to minimize these flaws. Apart from all our efforts in the revised manuscript, we have replaced Figure 6, by providing an improved quality (high-resolution image) of Figure 6. We are once again thankful for the reviewers’ critical remarks and suggestions.
On behalf of all co-authors: Prof. dr Hab Mariusz Pierzchala
Reviewer 2 Report
The manuscript has been improved; however, some minor changes are still required. Only some examples:
line 60: it shoul be "peri-implantation"
line 79" "shoul" or "must", not both of them
Introduction: decide "9th day" or "9D" and other days accordingly. Otherwise, the Reader is confused.
Line 174: it should be "Figures"
Western blot results: GAPDH seems to be inadequate internal control, as its level changes across the studied period.
Author Response
Response to Reviewer-2: In the revised manuscript, we have done extensive editing of the English language and style as per the IJMS guidelines to the authors. As Indicated by the reviewer, we have corrected lines 60, 79, 174, and two other corrections. Apart from all our efforts in the revised manuscript, we have replaced Figure 6, by providing an improved quality (high-resolution image) of Figure 6. We are once again thankful for the reviewers’ critical remarks and suggestions.
On behalf of all co-authors: Prof. dr Hab Mariusz Pierzchala